# Non-Invasive Assessment of Body Condition and Stress-Related Fecal Glucocorticoid Metabolite Concentrations in African Elephants (*Loxodonta africana*) Roaming in Fynbos Vegetation

**DOI:** 10.3390/ani10050814

**Published:** 2020-05-08

**Authors:** Elisabetta Carlin, Gabriella Teren, Andre Ganswindt

**Affiliations:** 1Mammal Research Institute, Department of Zoology and Entomology, Faculty of Natural and Agricultural Sciences, University of Pretoria, Pretoria 0028, South Africa; elisabettacarlin1985@gmail.com; 2Wildlife and Ecological Investments (WEI), Unit 20/21, Fountain Square, 136 Main Road, P.O. Box 3288, Somerset West 7130, South Africa; gabi.teren@wei.org.za

**Keywords:** fecal glucocorticoid metabolites, body condition scoring, nutrition, Cape Floral Kingdom, African elephant, drought

## Abstract

**Simple Summary:**

The Western Cape Province of South Africa is characterized by Fynbos vegetation. This unique endemic vegetation type belongs to the Cape Floral Kingdom, the smallest of the six floral kingdoms in the world, and only a few provincial, national and private game reserves in this area currently support populations of African elephants (*Loxodonta africana*). As a result, not much is known about the ability of elephants to thrive in such a unique environment. External factors such as a nutritionally poor diet can be perceived as a stressor by mammals, and establishing links between these two factors was the focus of this study. The results of monitoring individual stress-related hormone levels and body conditions suggest that elephants can obtain adequate nutrition from Fynbos vegetation.

**Abstract:**

Fynbos is a unique endemic vegetation type belonging to the Cape Floral Kingdom in the Western Cape Province of South Africa, representing the smallest of the six floral kingdoms in the world. Nowadays, only a few game reserves in this region support populations of African elephants (*Loxodonta africana*), and thus, little information exists regarding the suitability of the nutritionally poor Fynbos vegetation for these megaherbivores. Using already established non-invasive methods, the monitoring of individual body conditions and fecal glucocorticoid metabolite (fGCM) concentrations, as a measure of physiological stress, was performed to examine a herd of 13 elephants in a Western Cape Province Private Game Reserve, during two monitoring periods (April and June 2018), following a severe drought. The results indicate that overall median body condition scores (April and June: 3.0, range 2.0–3.0) and fGCM concentrations (April: 0.46 µg/g dry weight (DW), range 0.35–0.66 µg/g DW; June: 0.61 µg/g DW, range 0.22–1.06 µg/g DW) were comparable to those of other elephant populations previously studied utilizing the same techniques. These findings indicate that the individuals obtain sufficient nutrients from the surrounding Fynbos vegetation during the months monitored. However, a frequent assessment of body conditions and stress-associated fGCM concentrations in these animals would assist conservation management authorities and animal welfare practitioners in determining ways to manage this species in environments with comparably poorer nutritional vegetation.

## 1. Introduction

Previous studies reported the presence of considerable numbers of large herbivores (>20 kg) being resident as recently as 350 years ago in the Fynbos biome in the Western Cape Province of South Africa, including an estimated population of about 10,000 African bush elephants (*Loxodonta africana*) [1,2,3]. The Western Cape Province hosts the Cape Floristic Region (CFR), an 87,892 km^2^ area which is recognized as an important global hotspot of endemic biodiversity. The CFR is characterized by a number of different biomes, with the Fynbos Biome being the most prominent [4]. This environment typically has winter rainfall along the coast, while the mountain areas have a mild Mediterranean climate with mean annual temperatures of around 17 °C [4]. The biome is composed of Fynbos vegetation that covers around 19,227 km^2^ of the total area and includes species such as evergreen sclerophyllous shrubs, which rely on fire for regeneration and, due to the soil-poor nutrient contents, dominate the landscape, replacing trees [4].

Around the mid-17th century, after the European settlements became permanent in the Western and Eastern Cape Provinces, the abundance and distribution of large mammals changed rapidly [3,5,6]. Many large herbivore species, including the African elephant, faced local extinction in the CFR by 1800 due to trophy and ivory hunting and the eradication of “problem animals”, competing with settlers for natural resources [3,7]. This forced the last remaining populations to utilize less accessible areas such as the foothills of the Outeniqua and Tsitsikamma mountain ranges, around the Knysna Forest, or the dense-thicket areas known today as Addo Elephant Park [5,8]. The individuals of Knysna represent the southern-most elephant population in the world [9], but recent camera trap surveys demonstrated that the former Knysna population now comprises only one remnant female [10]. While historical records suggest some elephants were supported permanently in some fynbos habitats [2], fynbos vegetation may be of sub-optimal quality, given its low nutrient status [5,11,12]. This may be reflected in the overall low number of large animals occurring in the area compared to in savanna or grassland habitats [7]. African elephants are mixed feeders, with a selective and adaptive feeding regime that varies both spatially and seasonally from mostly woody browse to mostly grasses [13,14,15]. Fynbos shrublands are naturally lacking in both grass and tree elements, and recent investigations into opal phytoliths in dental calculus from historical free-roaming African elephants suggest that grazing on Restionaceae was important to the elephant diet in the CFR [16]. Recently, the burgeoning wildlife and tourism industry in the Western Cape has seen large herbivores such as the African elephant being reintroduced into newly created reserves in the CFR [17], with little information on how these animals are responding to this environment. Previous studies have shown how exposure to different environmental pressures, such as limited food availability, can be perceived as a stressor [18,19,20,21]. However, it is still not clear whether the low nutrient fynbos vegetation acts as a stressor for African elephants limited to such habitats and, if so, how these individuals cope with such a potential environmental pressure and maintain body conditions within acceptable limits [9]. In addition to the low nutrient status of fynbos shrublands, the CFR experienced the most severe drought in a century in 2015–2017 [22], further adding to environmental stress. 

For this study, stress is defined as any factor that threatens the homeostasis of an individual [23]. In attempting to restore homeostasis in situations of stress, the Central Nervous System activates, amongst others, the hypothalamic pituitary–adrenal axis, leading to an increase in glucocorticoid concentrations, as a countermeasure [24]. Although regarded as being adaptive in the short term, due to their role in basic energy acquisition, deposition and mobilization [25], a prolonged elevation of glucocorticoid concentrations may decrease individual fitness due to the suppression of immune functions, atrophy of tissues, reproduction failures, and behavioural and cognitional alterations [24,26,27]. Glucocorticoids or their metabolites can be quantified in different body matrices such as the blood, saliva, hair, fur, feathers, urine or feces [24,28]. Assessing glucocorticoid metabolite concentrations in feces (fGCMs) is a commonly used non-invasive approach when monitoring stress-associated alterations in hormone concentrations in wildlife, as the animal is generally not disturbed or affected by the collection of the fecal material, and thus, the measurement can be regarded as feedback-free [29]. For African elephants, a reliable enzyme-immunoassay (EIA) for monitoring fGCM concentrations has already been established [30], with fecal material collectable for up to 20 h post defecation [31].

Another option to assess the potential impact of low nutritional food quality in animals is to evaluate the individual body condition (BC), commonly used when managing domestic livestock and recognized as a standardized veterinary procedure [32,33,34]. The method of visually assessing subcutaneous fat in specific body regions using a scoring system has already been adapted for evaluating the BC of African elephants [35] and has been used to link BC alterations to reproductive activity and metabolic hormone concentrations in feces [26,36]. 

The aim of this study was to examine a herd of 13 free-roaming elephants within a reserve in the CFR characterized by Fynbos vegetation across a season following severe drought through (i) individual BC scores, using the methodology of Morfeld and colleagues [35], and (ii) fGCM concentrations, using the methodology of Ganswindt and colleagues [30].

## 2. Materials and Methods

### 2.1. Study Area and Animals

The study was conducted in a fenced private game reserve of 11,000 hectares located in the Western Cape of South Africa (34°04’51.78” South, 21°54’40.74” East), during the months of April and June 2018. Mean daily temperatures range between 11 and 21 °C and precipitation is generally a-seasonal, though rainfall occurs more frequently between autumn and spring (March–September). The long-term (1980–2019) average annual rainfall at the closest weather station at the Mossel Bay Lighthouse is 320 mm, and the total rainfall recorded in 2018 was 302 mm, with the majority occurring during the end of the wet season (September–October) [37]. Water is permanently available throughout the reserve via artificial waterholes. The site is dominated by fynbos vegetation, characterized by unique Mediterranean-type shrubland growing on nutrient poor soils, with about 10% of the site covered by old-agricultural grasslands of short lawn-forming *Cynodon dactylon*, which may provide elephants with some forage during winter [38]. The site presents also with localized woodland patches of native valley thicket and unpalatable alien acacia woodlands. The reserve hosts a range of free-roaming wildlife species reintroduced to the region, including the African elephant, lion (*Panthera leo*), African buffalo (*Syncerus caffer*), Southern giraffe (*Giraffa giraffa*) and white rhino (*Ceratotherium simum*), in addition to large herds of antelope. The grazers tend to congregate on the short lawn grasslands, while the elephants move around the reserve in search of suitable forage patches. The focal animal group consisted of 13 free-roaming elephants, introduced into the reserve between 2008 and 2014. The composition of the group included two adult bulls (34 and 29 years old), which generally roam together, and two matriarchal herds (α and β herd). The α herd is composed of two adult females (age unknown), four juveniles (3–7 years old) and one calf (1 year old), while the β herd is formed by only two adult females (both 26 years old) and two juveniles (5–6 years old). At the time of the study, all the elephants observed appeared to be in normal physical condition, and no sign of injury or any other form of ill health was recognized.

### 2.2. Determining Body Condition Score

For body condition scoring (BCS), a total of 78 suitable pictures (range: 3–6 pictures per individual per month), from the rear, the side and at a 45° angle from the body were taken. Two assessors evaluated the pictures independently and scored each individual for each month, using an established BCS index for African elephants [35]. The mean of the two scores was then assigned to each individual elephant for each of the two study months. This method is based on a numeric scoring system, ranging from 1/2 (emaciated/thin) to 3 (normal) to 4/5 (overweight/obese). Standardized photos of an animal are visually assessed by evaluating anatomical key areas (the ribs, the pelvic bone, the backbone and lumbar depression). Lower scores are indicators of poor health, malnutrition and low reproduction rates, while higher scores are associated with reproductive disorders, diabetes and other pathologies [35]. In this study, individual scores only ranged between 2 (thin) and 3 (normal), with an overall inter-rater reliability of 77%. 

### 2.3. Fecal Sample Collection

Fecal samples were collected opportunistically from as many individuals as possible during field work (33 fecal samples in total; mean: 1.1 samples, range: 0–4 samples per individual per field visit). Samples were collected within an hour of defecation [31], placed on ice immediately, and frozen at −20 °C at the end of each day in the field. To avoid cross-contamination with urine, soil or other feces present, material was removed from the middle of the boli of a dropping using sterile latex gloves. Collected material was kept frozen until reaching the Endocrine Research Laboratory (ERL) at the University of Pretoria for further processing.

### 2.4. Steroid Extraction and Analysis

Frozen feces were lyophilized, and the resulting dry matter was pulverized and sifted through a mesh to remove undigested fecal matter [39]. Subsequently, 0.050–0.055 g of fecal powder was extracted by vortexing for 15 min with 3 ML of 80% ethanol in water [26]. Extracts were measured for immunoreactive fGCM concentrations using an 11-oxoetiocholanalone enzyme immunoassay detecting fGCMs with a 5β-3α-ol-11-one structure, which have previously been shown to provide reliable information on adrenocortical function in African elephants [40,41,42]. Detailed assay characteristics, including full descriptions of the assay components and cross-reactivities, have been provided by Möstl and colleagues [43]. The sensitivity of the assay at 90% binding was 1.2 ng/g fecal dry weight (DW). Intraassay and interassay coefficients of variation (CV), as determined by the repeated measurement of high- and low-value quality controls, were 4.2% and 5.3% (intraassay CV) and 4.6% and 13.1% (interassay CV), respectively. All hormone analyses were performed at the Endocrine Research Laboratory, University of Pretoria, South Africa, and the results are expressed in µg/g dry weight (DW).

### 2.5. Data Analysis

Matching individual body condition scores of elephants monitored in April and June 2018 were compared using the Wilcoxon Signed Rank test. The individual fGCM concentrations of elephants monitored in April and June 2018 were compared using the Mann–Whitney Rank Sum test. When repeated values were available from the same animal during a month, a median value was used for analysis. To compare matching individual fGCM concentrations of elephants monitored during both months (n = 7), a paired t-test was utilized. Data subsets were tested a priori for normality using the Shapiro–Wilk test and, if required, for equal variance using the Brown–Forsythe test. 

Individual BC scores and individual median fGCM concentrations were correlated for April and June separately using the Pearson Product Moment Correlation test. All tests were two-tailed, with the level of significance set at 0.05. Sigma Plot version 12.5. (Systat Software, Inc., San Jose, CA, USA) was used for all statistical analyses.

## 3. Results

### 3.1. Body Condition Scores

Individual BCS ranged between 2.0 and 3.0 for the 13 focal animals during the first (April 2018) and second (June 2018) observational periods and did not change significantly between the two periods (W = 5; P = 0.375). Out of the 13 elephants observed, a lower BCS was assigned to three animals in April compared to in June, whereas the opposite was given for only one animal (Table 1). 

### 3.2. Fecal Glucocorticoid Metabolite (fGCM) Concentrations

Out of the 13 focal animals, fGCM concentrations could be determined for ten individuals for April and for nine individuals for June 2018, respectively. For seven animals, the fGCM concentrations could be obtained for both observational periods (Table 1). The overall individual median fGCM concentrations were significantly higher in June 2018 (April: 0.46 µg/g DW vs. June: 0.61 µg/g DW; T_10,9_ = 116.5, P = 0.034) and also showed a wider range (April: 0.35–0.66 µg/g DW vs. June: 0.22–1.06 µg/g DW). When comparing matching values, six out of the seven animals showed higher median fGCM concentrations in June compared to in April, but the median fGCM concentrations did not differ significantly (t_6_ = −1.78; P = 0.125). 

Individual median fGCM concentrations and BC scores were not correlated during the monitoring period in April (n = 10, r = −0.120, P = 0.741) or June 2018 (n = 9, r = −0.331, P = 0.384).

## 4. Discussion

In this study, variations in individual BCS and fGCM concentrations were investigated in a herd of 13 elephants, free-roaming in nutritionally poor fynbos vegetation. The overall median individual body condition scores were 3.0 for both sampling months (April and June) and only ranged from 2.5 to 3.0 for the majority of the elephants throughout the study. Furthermore, the fGCM concentrations determined over the same study period were comparable to the values determined in previous studies on elephants roaming in other parts of Africa [41,42,44,45]. Overall, for those animals where matched values were available, individual median fGCM concentrations were 33% higher in June compared to in April. The BC determined for the elephants in this study is in line with the findings of a previous study utilizing a photographic dataset of free-ranging African elephants in the Timbavati Private Game Reserve within Greater Kruger National Park, South Africa, where 72% of the scored animals (n = 57), received a BCS of 2 or 3, indicative of an ideal/normal body condition [35]. Our results showed no significant fluctuation occurring during the early and mid-wet season in individual BCS, contrary to the findings of Pokharel et al. [27], where differences in female Asian elephant BCS between the early and mid-wet season were observed and linked to extended drought conditions during the time of the study. A most likely explanation for the lack of the early and mid-wet seasonal-related changes in BCS in our study animals is missing data for the opposite peak during the study period. On an individual level, Bull β1 was the only monitored individual to exhibit a BCS of 2.0, for both the April and June 2018 sampling periods. A possible explanation for the comparatively low BC of this bull could be that he entered into the state of musth, a temporal condition of increased cortisol levels and sexual activity associated with a number of physical, physiological and behavioural characteristics including decreased feeding activity, which leads to a progressive decline in body condition [36]. However, no signs of musth were noted in Bull β1 during the entire monitoring period. Thus, additional observations of this individual would be necessary to identify potential causes impacting on the bulls’ BCS. The only other individual exhibiting a BCS of 2.0 during the study was Female β2, a mother of a not-completely-weaned five-year-old juvenile. This female, unlike other lactating females that maintained a BCS between 2.5 and 3.0 for the whole study period, showed an increase in BCS from 2.0 in April to 2.5 in June. Mature females can have higher nutritional requirements then adult males in general, due to the energy-demanding status of pregnancy or lactation, and thus become more vulnerable to physiological or nutritional stress [46,47,48,49,50]. In such conditions, cows make their body fat available to balance lactation requirements, and this could presumably be reflected in lower BCS, as seen in similar studies on wild populations [51,52]. The slowly increasing body conditions of Female β2, however, show that this individual is able to sustain herself and provide enough nutriment to her youngster. Unfortunately, we did not manage to get an fGCM value for Female β2 for June to be compared with the average fGCM concentration determined for this female for April. Therefore, any discussion about a potential modulating effect of elevated glucocorticoid concentrations to support existing energy demands, as seen in other studies [51,52], would be speculative.

The fGCM concentrations determined in this study appear to be comparable to the baseline fGCM values determined in previous studies on elephants using the same non-invasive approach to determine stress-related hormone levels [26,40,41,45]. Overall individual median fGCM concentrations differed by 33% between the early and mid-wet season. Part of this variation could be related to changes in the estrus cycle of adult females, as the higher fGCM concentrations occurring after the follicular phase and the two monitoring periods (April and June) are far enough apart to represent two different phases of a cycle [53]. Furthermore, the moderate increase in the fGCM concentration in the mid-wet season could be linked, to some extent, to the exceptional dry conditions over the winter rainfall season registered in the region between 2015 and 2018 [22,54,55], which also impacted agriculture, livestock production and the water supply [55,56,57]. While the artificial waterholes maintained water for drinking, the perennial lawn grasslands dominated by *Cynodon dactylon*, which may provide important winter forage for elephants in the Eastern Cape [38], remained brown during our sampling period, with as much dead grass cover as living [58]. The elephants did not have access to supplementary forage. The long-term consequences of these prolonged dry conditions on the vegetation should be investigated further, to unravel potential shifts in seasonal variations and their relation to stress-related responses in wildlife. Bull β1 was one of the two individuals exhibiting comparatively high fGCM concentrations (see Ganswindt et al. [26], for comparison), also showing also a distinct 50% decrease in fGCM concentrations from April (0.98 µg/g DW) to June (0.49 µg/g DW).This temporal rise in fGCM concentrations seen in April could be the result of various factors, such as normal increased physiological activity [59], competition for access to females [60], or the result of disease or injury [26]. However, no indication of reproductive competition, disease or injury were noted during the study period, so we cannot certainly ascribe this variation to stress conditions. The other elephant showing comparatively high fGCM concentrations was Juvenile α3 (June: 1.06 µg/g DW), but due to the lack of a comparable data point for this animal for April, we can only speculate if this individual perceived a stressor over a prolonged period or reacted to a specific stressor occurring in June. Thus, a more comprehensive, ideally longitudinal, dataset covering multiple seasons would be required to pinpoint energy-demanding and other factors perceived as stressors and thus reflected in elevated fGCM concentrations in these elephants. In previous studies of this nature, it was described how annual fGCM concentration measurements were an objective indication of wellness in elephants, as well as how annual fluctuations in body condition related to climatic variation [27,60]. Thus, longitudinal physiological data in combination with BCS, together with additional comprehensive measurements of ecological conditions, could also allow a better understanding of the overall coping mechanisms these elephants use to thrive in such environments.

BC scores and fGCM concentrations were not correlated in our study animals. This is in line with the findings of Kumar et al. [61], who studied this type of relationship in 12 Asian elephants housed in three zoos in southern India. Contrarily, a number of previous studies on mammals such as wild rabbits (*Oryctolagus cuniculus*), Steller sea lions (*Eumetopias jubatus*), and African and Asian elephants (*Elephas maximus*) [24,25,62,63] demonstrated a negative relationship between fGCM concentrations and BCS. A potential explanation for the absence of any relationship between BCS and fGCM concentrations in our study animals could be the limited number of data points available in combination with a lack of any distinct alteration of BCS during the two months of study. 

## 5. Conclusions

Our study provides evidence that the monitored elephants living in the Fynbos biome maintain a BCS within acceptable limits and show stress-related fGCM concentrations comparable to the baseline fGCM values determined for other African elephant populations, ultimately suggesting that elephants can exist in reserves in CFR areas. However, in our opinion, reserve management would benefit from continuous monitoring over an extended period of time to truly evaluate the conservation and welfare implications.

## Figures and Tables

**Table 1 animals-10-00814-t001:** Individual body condition scoring (BCS) and fecal glucocorticoid metabolite (fGCM) concentrations of 13 elephants assessed in April and June 2018. BCS was done on a point scale ranging from 1 to 5. Individual median fGCM concentrations were calculated if the individual sample size was n > 1 for a respective month (number of individual samples collected and the values or range are given in brackets below).

Elephant ID	BCS April 2018	BCS June 2018	Median fGCM (µg/g DW) April 2018	Median fGCM (µg/g DW) June 2018
Adult F. α 1	2.5	2.5	0.46 (n = 1)	0.52 (n = 2; 0.46, 0.59)
Adult F. α 2	3	2.5	0.35 (n = 3; range 0.32–0.44)	0.65 (n = 4; 0.59–0.92)
Juvenile α 1	3	3	-	-
Juvenile α 2	2.5	3	0.40 (n = 1)	0.61 (n = 1)
Juvenile α 3	3	3	-	1.06 (n = 2; 1.36,0.76)
Juvenile α 4	3	3	-	0.51 (n = 1)
Calf α 1	3	3	0.66 (n = 1)	-
Adult M. β 1	2	2	0.49 (n = 2; 0.37,0.61)	0.98 (n = 1)
Adult M. β 2	3	3	0.47 (n = 2; 0.44, 0.50)	0.22 (n = 2; 0.22, 0.21)
Adult F. β 1	3	3	0.40 (n = 2; 0,43, 037)	0.49 (n = 2; 0,50, 0,66)
Adult F. β 2	2	2.5	0.46 (n = 2; 0.35, 0.57)	-
Juvenile β 1	2.5	3	0.58 (n = 1)	0.76 (n = 1)
Juvenile β 2	3	3	0.36 (n = 3; range 0.37–0.36)	-
Overall median	3	3	0.46 ^a^	0.61 ^b^

Different super-scripts (^a,b^) indicate statistically significant differences between groups.

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
