# Peer review of "Non-Invasive Assessment of Body Condition and Stress-Related Fecal Glucocorticoid Metabolite Concentrations in African Elephants (Loxodonta africana) Roaming in Fynbos Vegetation"

_animals, 2020, doi:10.3390/ani10050814_

Round 1
Reviewer 1 Report
Manuscript ID: animals-771135
Title: Non-invasive assessment of body condition and stress-related faecal glucocorticoid metabolite concentrations in African elephants (Loxodonta africana) roaming in Fynbos vegetation
This study assesses the body condition and fGCM concentration of African elephants in a reserve where habitat is composed of fynbos - shrubland or heathland vegetation - which is not considered to be prime habitat for this species. Despite the reported lower nutritional value of this habitat, neither body condition or faecal glucocorticoid metabolites were found to be substantially different to those previously reported in other populations, suggesting that this non-traditional habitat my be able to sustain at least a small (and breeding) population. As habitat for African elephants is under competition for other uses, the potential for expanding their range could be useful for species management. Although the number of samples collected was relatively low per individual, and data collection periods were only separated by a month, making seasonal changes more difficult to assess, this study does provide interesting baseline information for this population. As the authors note, more longitudinal analyses would be useful both to understand the long-term impacts on the African elephant population and on the habitat itself. The manuscript is well written overall but could benefit from a couple of clarifications prior to publication - my suggestions are given by line below.
Line 30: Should read ‘nutritionally’
Line 33: ‘was’ should be ‘were’
Line 35: Not sure that ‘Results’ should be bold as this is part of the abstract (unless this is journal formatting?)
Line 37: ‘within previously published acceptable limits’ - this is difficult because of lab to lab variability - so it might be better to state that these were comparable to other populations studied previously (i.e. comparison to data from the same lab)
Line 47: Please clarify your meaning here - I assume that you mean they were present there as recently as 350 years ago (and aren't now?), but this could also be read as they were only present 350 years ago (and not before).
Line 52: It would be useful to briefly explain what the Fynbos biome is here
Line 60: Please delete ‘.’ after reference
Line 82: It should be ‘glucocorticoids or their metabolites’, because glucocorticoids are plural
Line 115: Please give approximate age of the calf (or your criteria for a calf being e.g. less than 1 year of age, to differentiate from the 2-year-old juvenile)
Line 121: It would be useful to provide a short summary of this method (or at least what the range of potential scores are), so that readers don't have to go find the cited reference to be able to interpret if the study elephants are in good vs poor condition.
Lines 121-122: Were any criteria used for ensuring the mean was appropriate - i.e. did observers have to agree to within certain percentage for scores to be used? Inter-observer reliability is typically reported in scenarios like this.
Line 133: ‘were’ should be ‘was’
Line 162: Table legend - so that table can stand alone, it would be useful to include what point scale the BCS was scored on. It would also be informative to include N for faecal samples for each individual, and also the range in fGCM concentrations where more than one sample was collected - this will help the reader to interpret whether any more extreme values might be outlying points that could be associated with acute adrenal activation, or more consistent concentrations within an individual.
Table 1: More of a formatting comment - but I suggest you use Adult F and Adult M, so that these can fit on one line. Also please check that all fGCM concentrations in the table are given to the same number of decimal places
Lines 176-186: Section 3.1 is repeated.
Line 193: Please delete ‘a’ after Africa
Line 193: Please check references are correct - #42 does not contain fGCM data
Line 199: Data were collected only a month apart - I’m not sure that this is enough of a difference to state that there were no seasonal changes, unless environmental conditions are vastly different between April and June. There is presumably a lag in body condition response to changes in environmental conditions - changes in rainfall take some time to impact vegetation growth and nutritional quality, and BC takes time to increase or decline as food intake or quality changes. I suggest that more details are needed about conditions in April vs June to allow the reader to understand whether seasonal changes would be expected, and a little more discussion would be beneficial also - if these are the two most extreme months then that should be discussed, if they are not, then do the authors anticipate body condition to become more compromised during a different time of the year?
Line 221: Again, please check references - #42 does not contain fGCM data
Line 222: Should read ‘differed’
Line 222: The authors here mention a difference in fGCM of 33% between early and late wet season, but in the methods they describe rainfall as being a-seasonal, with precipitation more frequent from March - September. Please edit the discussion to reflect that the data periods were early and mid wet season, and what implications this may have for the observed body condition and fGCM results.
Line 248: Should read ‘months’
Line 252: Should read ‘ultimately’
Line 255: Suggest changing ‘benefits’ to ‘implications’
Author Response
Comments and Suggestions for Authors
Reviewer 1
Manuscript ID: animals-771135
Title: Non-invasive assessment of body condition and stress-related faecal glucocorticoid metabolite concentrations in African elephants (Loxodonta africana) roaming in Fynbos vegetation
This study assesses the body condition and fGCM concentration of African elephants in a reserve where habitat is composed of fynbos - shrubland or heathland vegetation - which is not considered to be prime habitat for this species. Despite the reported lower nutritional value of this habitat, neither body condition or faecal glucocorticoid metabolites were found to be substantially different to those previously reported in other populations, suggesting that this non-traditional habitat may be able to sustain at least a small (and breeding) population. As habitat for African elephants is under competition for other uses, the potential for expanding their range could be useful for species management. Although the number of samples collected was relatively low per individual, and data collection periods were only separated by a month, making seasonal changes more difficult to assess, this study does provide interesting baseline information for this population. As the authors note, more longitudinal analyses would be useful both to understand the long-term impacts on the African elephant population and on the habitat itself. The manuscript is well written overall but could benefit from a couple of clarifications prior to publication - my suggestions are given by line below.
We thank the reviewer for the constructive and encouraging comments.
Line 30: Should read ‘nutritionally’
We have changed this as suggested (line 30 in the revised MS).
Line 33: ‘was’ should be ‘were’
Has been changed accordingly (line 33 in the revised MS).
Line 35: Not sure that ‘Results’ should be bold as this is part of the abstract (unless this is journal formatting?)
We adjusted the format accordingly (line 36 in the revised MS).
Line 37: ‘within previously published acceptable limits’ - this is difficult because of lab to lab variability - so it might be better to state that these were comparable to other populations studied previously (i.e. comparison to data from the same lab)
Thank you for pointing this out, we rephrased the sentence now reading ‘…were comparable to those of other elephant populations previously studied utilizing the same techniques.’ (line 39 in the revised MS).
Line 47: Please clarify your meaning here - I assume that you mean they were present there as recently as 350 years ago (and aren't now?), but this could also be read as they were only present 350 years ago (and not before).
We apologize for the misleading statement and changed it to ‘…the presence of considerable numbers of large herbivores (> 20 kg) being resident as recently as 350 years ago…’ (Line 49 in the revised MS)
Line 52: It would be useful to briefly explain what the Fynbos biome is here
We added brief explanation as suggested (line 54 in the revised MS)
Line 60: Please delete ‘.’ after reference
This was deleted (line 67 in the revised MS).
Line 82: It should be ‘glucocorticoids or their metabolites’, because glucocorticoids are plural
We apologize for this oversight and changed it accordingly (line 89 in the revised MS).
Line 115: Please give approximate age of the calf (or your criteria for a calf being e.g. less than 1 year of age, to differentiate from the 2-year-old juvenile)
We added information regarding the age of the juveniles and calf (line 132 in the revised MS).
Line 121: It would be useful to provide a short summary of this method (or at least what the range of potential scores are), so that readers don't have to go find the cited reference to be able to interpret if the study elephants are in good vs poor condition.
Thank you for pointing this out. We added respective information to the MS (line 142 in the revised MS).
Lines 121-122: Were any criteria used for ensuring the mean was appropriate - i.e. did observers have to agree to within certain percentage for scores to be used? Inter-observer reliability is typically reported in scenarios like this.
As all individual scores given ranged only between a BCS of 2 and 3, we believe that the mean out of two individually given scores as the finally assigned score for an individual for a given month is a suitable approach. As suggested, we also added the following information regarding the inter-rater reliability; ‘In this study, individual scores only ranged between 2 (thin) and 3 (normal), with an overall inter-rater reliability of 77%.’ (line 147 in the revised MS).
Line 133: ‘were’ should be ‘was’
This was changed (line 150 in the revised MS).
Line 162: Table legend - so that table can stand alone, it would be useful to include what point scale the BCS was scored on. It would also be informative to include N for faecal samples for each individual, and also the range in fGCM concentrations where more than one sample was collected - this will help the reader to interpret whether any more extreme values might be outlying points that could be associated with acute adrenal activation, or more consistent concentrations within an individual.
We thanks the reviewer for these suggestions and adjusted the table and its legend accordingly.
Table 1: More of a formatting comment - but I suggest you use Adult F and Adult M, so that these can fit on one line. Also please check that all fGCM concentrations in the table are given to the same number of decimal places
We have changed this as suggested (Table 1 in the revised MS).
Lines 176-186: Section 3.1 is repeated.
We apologize for this formatting mistake and deleted the repeated heading (line192-202 in the revised MS).
Line 193: Please delete ‘a’ after Africa
This was deleted (line 210 in the revised MS).
Line 193: Please check references are correct - #42 does not contain fGCM data
We apologize for this oversight and corrected the reference (line 209 in the revised MS).
Line 199: Data were collected only a month apart - I’m not sure that this is enough of a difference to state that there were no seasonal changes, unless environmental conditions are vastly different between April and June. There is presumably a lag in body condition response to changes in environmental conditions - changes in rainfall take some time to impact vegetation growth and nutritional quality, and BC takes time to increase or decline as food intake or quality changes. I suggest that more details are needed about conditions in April vs June to allow the reader to understand whether seasonal changes would be expected, and a little more discussion would be beneficial also - if these are the two most extreme months then that should be discussed, if they are not, then do the authors anticipate body condition to become more compromised during a different time of the year?
We thank the reviewer for the suggestion and acknowledge that the two periods used for sample collection will not be sufficient to evaluate seasonal changes. We therefore aligned our approach in this regard, now referring to changes across a season following severe drought throughout the manuscript. To underline the conditions experienced during the study, we added information regarding the limited rainfall seen in the year of study as well as what the region usually experienced over the last decades (line 117-118 in the revised Ms).
Line 221: Again, please check references - #42 does not contain fGCM data l.232
We apologize for this oversight and corrected the reference (line 244 in the revised MS).
Line 222: Should read ‘differed’
Thanks, we corrected this accordingly (line 243 in the revised MS).
Line 222: The authors here mention a difference in fGCM of 33% between early and late wet season, but in the methods they describe rainfall as being a-seasonal, with precipitation more frequent from March - September. Please edit the discussion to reflect that the data periods were early and mid-wet season, and what implications this may have for the observed body condition and fGCM results.
We thank the reviewer for pointing that out. We have changed this as suggested (line 244 of the revised manuscript).
Line 248: Should read ‘months’
This was changed (line 281)
Line 252: Should read ‘ultimately’
This was changed (line 285)
Line 255: Suggest changing ‘benefits’ to ‘implications’
We have changed this as suggested (line 288)
Reviewer 2 Report
Overall a very interesting article, which takes up a current issue. There are a few things that require attention.
Keywords: I suggest you delete “non-invasive hormone monitoring”
LINE 63 and 74-75: “..fynbos vegetation may be of non-optimal quality given its low nutrient status”. “…However, it is still not clear whether the low nutrient fynbos vegetation acts “. You aimed to assess the potential impact of low nutritional food quality in animals. Why did you not analyze the pasture?
Line 80-82: “However, when a stressor is perceived over a prolonged period, it can also cause suppression of immune functions, atrophy of tissues, reproduction failures as well as behavioural and cognitional alterations [23–25].” It would be useful here to add some information about the link between the stress hormone levels and BCS (e.g the role of glucocorticoid hormones in energy mobilization) rather than the correlation with blood parameters that you didn’t analyze.
Line 104: Why April and June? It would have been interesting to have more data over a longer period. I suggest you add just a few information about the South African rainy and dry seasons to improve clarity to anyone less familiar with this topic.
Line 118: it would be useful here to add a sentence about the health status of the elephants included in this dataset – were all individuals considered to be clinically normal at the time of sample collection?
Line 124: “Faecal samples were collected opportunistically from as many individuals as possible during field work (42 faecal samples in total; mean: 1.1 samples, range: 0-4 samples per individual per field visit)” Please clarify this point. As animals develop a circadian rhythm of glucocorticoids concentrations it would be important if you considered or standardized time of sampling in all animals and preclude the influence of daytime on concentrations.
Line 176-186: please delete 3.1. Faecal glucocorticoid metabolite (fGCM) concentrations
Line 252: “ultimately suggesting that elephants can be supported on larger reserves in CFR areas.” Please could you clarify this?
Author Response
Reviewer 2
Comments and Suggestions for Authors
Overall a very interesting article, which takes up a current issue. There are a few things that require attention.
We thank the reviewer for the encouraging statement.
Keywords: I suggest you delete “non-invasive hormone monitoring”
The respective keyword has been deleted.
Line 63 and 74-75: “...fynbos vegetation may be of non-optimal quality given its low nutrient status”. “…However, it is still not clear whether the low nutrient fynbos vegetation acts “. You aimed to assess the potential impact of low nutritional food quality in animals. Why did you not analyze the pasture?
We agree with the reviewer that further information regarding the nutritional content of the pasture would be highly beneficial, but as this study was an honours research project, with some distinct time and financial restrictions, we unfortunately couldn’t add another analytical component to the study. To alleviate this shortcoming, we added some basic information regarding the nutritional value of the grassland, based on a report provided by Wildlife and Ecological Investments (WEI) (line 122-124 in the revised MS). However, this aspect could definitively be a relevant topic for some subsequent research, as further studies on feeding ecology are currently in the planning stage by WEI.
Line 80-82: “However, when a stressor is perceived over a prolonged period, it can also cause suppression of immune functions, atrophy of tissues, reproduction failures as well as behavioural and cognitional alterations [23–25].” It would be useful here to add some information about the link between the stress hormone levels and BCS (e.g the role of glucocorticoid hormones in energy mobilization) rather than the correlation with blood parameters that you didn’t analyze. (l87-89)
We thank the reviewer for this suggestion and adjusted the statement accordingly, now reading: ‘…as a countermeasure [24]. Although regarded as being adaptive in the short term, due to their role in basic energy acquisition, deposition, and mobilization [25], a prolonged elevation of glucocorticoid concentrations may decrease individual fitness due to the suppression of immune functions,…’ (lines 89-92 in the revised MS).
Line 104: Why April and June? It would have been interesting to have more data over a longer period. I suggest you add just a few information about the South African rainy and dry seasons to improve clarity to anyone less familiar with this topic.
We agree with the reviewer that additional data collected over a longer period would be highly beneficial. We tried to alleviate this shortcoming by adding some information regarding the limited rainfall the area experienced in the year of study in comparison to previous years, but as this study was an honours research project, with some distinct time restrictions, we unfortunately couldn’t extent our fieldwork period. To further address this point, we conclude in the manuscript that monitoring over a longer period would be beneficial to reliably evaluate the welfare of the elephants population and the habitat. As already mentioned in response to a similar point raised by reviewer 1, we also changes our focus with regards to the two months monitored, now emphasising changes across a season following severe drought instead of evaluating seasonal changes.
Line 118: it would be useful here to add a sentence about the health status of the elephants included in this dataset – were all individuals considered to be clinically normal at the time of sample collection?
Thank you for this suggestion, a respective statement was added (line 134 in the revised manuscript.
Line 124: “Faecal samples were collected opportunistically from as many individuals as possible during field work (42 faecal samples in total; mean: 1.1 samples, range: 0-4 samples per individual per field visit)” Please clarify this point. As animals develop a circadian rhythm of glucocorticoids concentrations it would be important if you considered or standardized time of sampling in all animals and preclude the influence of daytime on concentrations.
We thank the reviewer for raising that point. Due to the logistical challenges of first finding the elephants at the reserve and then waiting for an opportunity to collect respective faecal material, sample collection took place on a daily basis from 7 a.m. to 7 p.m. during the two months indicated, to maximise success. As circulating hormone levels in the faeces are integrated over a certain period, faecal hormone metabolite levels reflect the production rate rather than a point in time, so the cumulative secretion and elimination of hormones, over at least several hours. As a consequence, unlike blood samples, faecal samples are less affected by episodic fluctuations or the pulsatility of hormone secretion (Touma and Palme 2005, Ann. N.Y. Acad. Sci. 1046 54–74). As we collected over a 12 hour time window, we thus believe that our data set allows a reliable comparison.
Line 176-186: please delete 3.1. Faecal glucocorticoid metabolite (fGCM) concentrations
We apologize for this formatting mistake and deleted the repeated heading (line 192-202 in the revised MS).
Line 252: “ultimately suggesting that elephants can be supported on larger reserves in CFR areas.” Please could you clarify this?
We apologize as the word ‘larger’ was a mistake and has been omitted (line 286 in the revised MS). Here we would like to point out that our results indicate that food availability and nutrient access in Fynbos vegetation meet the physiological needs of the study animals, so we suggest that elephants can exist in such a unique environment. We rephrased the statement accordingly for clarification.
Reviewer 3 Report
Overall: This study provides basic physiological information that can be built upon over time especially if these animals are continually monitored as is the hope of this reviewer. Longitudinal dataset on these animals and unique ecosystem would be very beneficial to the management and conservation of this species. Overall, I recommend this article for publication in Animals, with minor, but important, clarifications/adjustments to the methods and discussion especially. In general, I believe mention of “different seasons” or “across seasons” should be removed from the discussion as the samples and photographs were taken in the middle of the same season. Reinterpretation of the results will need to be incorporated throughout. Otherwise, why April and June (middle of rainy season March – September) are considered two different seasons should be clearly explained in the context of CFR.
Line 63: consider rewording to “…fynbos vegetation may be of sub-optimal quality, given its low nutrient content”
Line 75-76: Animals, particularly those in seasonal climates with varying food availability can maintain appropriate homeostasis over a range of body conditions. The goals is not one body condition, especially in African elephants that have shown the ability to drop and gain body condition over several seasons, and are still able to reproduce and raise calves. Consider re-wording to something along the lines of ”how these individuals cope with such a potential environmental pressure and maintain body condition within a healthy range.”
Line 80: The word “counteract” is a bit clunky in this sentence, consider rewording, maybe the authors meant “countermeasure”?
Additional explanation of the physiological response to an increase in cortisol is needed and appropriate. Especially given that cortisol functions as metabolic factor increasing glucose availability to the body, and this papers discussion focuses on the relationship of fGCM and body condition, clearing indicating there is a metabolic association between the two, yet this critical role of cortisol in maintaining homeostasis is not even mentioned.
Line 104: Authors need to clearly state that April and June are within the same season in CFR. It is not mentioned until the discussion and should be clearly pointed out in the methods to ease interpretation.
Also, why were these two months in particular chosen for the study design? Ease of tracking animals? They are more active? Why the start and end of the same season? Why not the end of two different seasons?
Line 109: what type of grassland is present in the old-agricultural areas? And how often is it known that these elephant visit these areas? Some of these grasses can be quite high in nutritional content. Mention of distance to closest human settlement is needed as elephant can obtain additional nutritional support from crops, and knowledge of raiding activity should be declared even if it is “we don’t know if these elephants raid”.
Line 119: Was inter-assessor reliability undertaken for the photographs? Agreement between the inter-assessor assessments should be reported and included in this paper. Those values not in agreement should be re-evaluated or not included in further analyses. A good explanation of this process is given in Morfeld et al. 2016 (doi: 10.1371/journal.pone.0155146).
Line 126: Further clarification of how samples were collected is needed. Generally, sub samples of several bolus are taken to avoid hormone “hot spots”. Hormone metabolites are not evenly distributed across an entire fecal sample, especially animals that defecate in bolus form and where one bolus is a part of a whole. Sub-sampling several boluses to get a “homogenous” sub-sample is best practice. As I understand it, sub-samples were taken from only one bolus per elephant. If that is the case it should be clearly stated or otherwise clarified further.
Was GPS location of the samples recorded? A map of the CFR area would be good to include with home range for each herd depicted (if known), with particular interest to their proximity to artificial water sources mentioned and human settlements.
Table 1: Inclusion of number of faecal samples analyzed per individual is needed in the table for proper interpretation of the results given the low number of samples and the wide range of samples per individual (0-4). Indicating that some values reported are one-off samples while others are the median, but the title of the table says median for all which is not the case for the n=1 individuals.
Additionally, adding an asterix or some other indicator of significance to table 1 between the April and June values would be helpful to the reader in interpreting the table.
Section 3.1 is repeated in the manuscript reviewed.
Line 188-189: It is known what these elephants eat? How many different vegetation types do they consume and do they consume agricultural crops? An additional clarifying statement regarding what is known about what they consume should be included, otherwise a statement that it is not known and they could be supplementing with agricultural crops should be mentioned.
Lines 193-194: Need to temper this assertion as it could only be verified for six individuals based on very low number of samples. Something along the lines of: "for the individuals where matched values were available, median fGCM concentrations were 33% higher in June compared to April".
Most the animals observed are females, and it is known that fGCM fluctuate with the ovarian cycle in elephants with higher concentrations occurring during the follicular phase (see Fanson et al. 2014; doi: 10.1530/EC-14-0025). April and June are far enough apart to potentially be in two different phases of the ovarian cycle and could explain the differences seen in the concentrations. Although knowledge of the ovarian cycle for each individual elephant is beyond the scope of this paper, it needs to be mentioned and discussed as possible explanation.
Line198-199: the statement “our results showed no seasonal fluctuation in individual BCS…” is confusing because to my understanding samples and pictures were taken in the middle of the same rainy season (March – September 2018). Comparing to studies that looked at BCS across seasons is not appropriate for interpretation of the results.
There’s are also no “seasonal related changes in BCS” (line 201-202) because there are no different seasons in this study design that I can find. Alternatively, explaining that within season there is no difference makes more sense, and that there is a need to investigate this relationship across seasons in future studies.
Cortisol also increases during musth in bulls, some incorporation of this in the discussion could be useful in explaining the results.
Lines 215-216: From the explanation of the lactating female with BCS 2.0 it does not seem to me that she is under nutritional stress. She is able to provide nutritional support for her calf and herself while increasing BCS in three months, she does not appear to be compromised in any way. Knowing which one she is in terms of fGCM would be useful in this discussion as cortisol is primarily a metabolic hormone. Discussion of cortisols role in maintaining metabolic homeostasis during lactation is needed and would round out the discussion nicely. As much broad perspectives as needed should be taken given the limited data available.
Lines 222-223: How much difference was there in rain during the same season to account for this interpretation? Especially given that artificial water sources are also present in this area. This is over interpreting data in my opinion and should be re-examined.
Lines 226: Discussion of an exceptionally dry conditions over the winter rainfall season needs to include discussion of the artificial water sources that are present in this area. How often are the elephants known to visit these sites? Wouldn’t the dry conditions be offset by the artificial water sources? How much impact do they have on the ecology of the area?
Lines 233-236: Discussion of the other potential sources for an increase in fGCM should include the possibility of increased activity of individual B1. With more activity cortisol will increase and would not necessarily be the result of “stress induced” increases in the measured fGCM. An animal that can modulate cortisol secretion to better cope in a changing environment is able to adapt and respond to those changes and would be maintaining homeostasis. A broader view is needed in this discussion overall. The animals appear to be coping well, which is great, more exploration of why that would be and what facilities their persistence in this area would be good alternatives to explore rather then a singular and very simplistic and outdated “stress” perspective.
Lines 243-244: Hard to compare free roaming animals to animals held in zoos that are generally overfed and have reduced activity in comparison, with very different environmental and psychological stressors to contend with. If you need examples of other studies that looked at BCS and fGCM there is a wealth of information available in other species such as bears (omnivores and I think still appropriate to use) and ungulates. Those would be better comparisons for a general discussion of how associations between BCS and fGCM can be used to optimize conservation strategies and the challenges with making these associations in the face of complex dietary patterns and seasonal variations.
Lines 248: typo – correct to moths to months. And wasn’t it three months of study?
Author Response
Reviewer 3
Comments and Suggestions for Authors
Overall: This study provides basic physiological information that can be built upon over time especially if these animals are continually monitored as is the hope of this reviewer. Longitudinal dataset on these animals and unique ecosystem would be very beneficial to the management and conservation of this species. Overall, I recommend this article for publication in Animals, with minor, but important, clarifications/adjustments to the methods and discussion especially. In general, I believe mention of “different seasons” or “across seasons” should be removed from the discussion as the samples and photographs were taken in the middle of the same season. Reinterpretation of the results will need to be incorporated throughout. Otherwise, why April and June (middle of rainy season March – September) are considered two different seasons should be clearly explained in the context of CFR.
We thank the reviewer for the detailed and constructive comments and hope that our revised version sufficiently addresses the points raised.
Line 63: consider rewording to “…fynbos vegetation may be of sub-optimal quality, given its low nutrient content”
Thank you for your comment, we agree. It was changed (line 69 in the revised MS)
Line 75-76: Animals, particularly those in seasonal climates with varying food availability can maintain appropriate homeostasis over a range of body conditions. The goals is not one body condition, especially in African elephants that have shown the ability to drop and gain body condition over several seasons, and are still able to reproduce and raise calves. Consider re-wording to something along the lines of ”how these individuals cope with such a potential environmental pressure and maintain body condition within a healthy range.”
We agree with this suggestion and re-phrased accordingly (line 81-82 in the revised MS).
Line 80: The word “counteract” is a bit clunky in this sentence, consider rewording, maybe the authors meant “countermeasure”?
We agree and adjusted the wording accordingly (line 88 in the revised MS).
Additional explanation of the physiological response to an increase in cortisol is needed and appropriate. Especially given that cortisol functions as metabolic factor increasing glucose availability to the body, and this papers discussion focuses on the relationship of fGCM and body condition, clearing indicating there is a metabolic association between the two, yet this critical role of cortisol in maintaining homeostasis is not even mentioned.
We thank the reviewer for pointing out this shortcoming and in accordance to a similar comment from reviewer 2, we added additional information reading the role of GCs as metabolic factors increasing energy availability.
‘…as a counteract measure [23]. Although regarded as been adaptive in the short term, due to their role in basic energy acquisition, deposition, and mobilization [59], a prolonged elevation of glucocorticoid concentrations may decrease individual fitness due to the suppression of immune functions…’ (lines 88-91 in the revised MS).
Line 104: Authors need to clearly state that April and June are within the same season in CFR. It is not mentioned until the discussion and should be clearly pointed out in the methods to ease interpretation. Also, why were these two months in particular chosen for the study design? Ease of tracking animals? They are more active? Why the start and end of the same season? Why not the end of two different seasons?
We agree with the reviewer that a study design including the end of the two different seasons for sample collection would have been ideal. As explained above, in line with a similar comment from reviewer 2, we tried to alleviate this shortcoming by adding some information regarding the limited rainfall the area experienced in the year of study in comparison to previous years, and now also emphasising changes across a season following severe drought instead of evaluating seasonal changes. Unfortunately, we couldn’t extent the fieldwork period for this study to include the ideal time windows, as this study was an honours research project, with some distinct time restrictions. We further underline this shortcoming by concluding in the manuscript that, a prolonged monitoring would be beneficial to reliably evaluate the welfare of the elephant population and the habitat.
Line 109: what type of grassland is present in the old-agricultural areas? And how often is it known that these elephant visit these areas? Some of these grasses can be quite high in nutritional content. Mention of distance to closest human settlement is needed as elephant can obtain additional nutritional support from crops, and knowledge of raiding activity should be declared even if it is “we don’t know if these elephants raid”
We thank the reviewer for those thoughts. Unfortunately we don’t have any information regarding time budget or movement pattern and thus can’t say anything about the actual presence of the elephants in these areas. To address the point of additional nutritional support through crop raiding, we added information that the elephants could not access crops of nearby areas due to fencing (line 112). We also added a sentence that elephants were not supplementary fed (line 254 in the revised MS). Regarding the nutritional value of the grassland, we added some information provided by Wildlife and Ecological Investments (WEI) via an unpublished monitoring report were it says that ‘…about 10% of the site covered by old-agricultural grasslands of short lawn-forming Cynodon dactylon, that may provide elephants with some forage during winter [60].’ (line 122-123 in the revised MS).
Line 119: Was inter-assessor reliability undertaken for the photographs? Agreement between the inter-assessor assessments should be reported and included in this paper. Those values not in agreement should be re-evaluated or not included in further analyses. A good explanation of this process is given in Morfeld et al. 2016 (doi: 10.1371/journal.pone.0155146).
We thank the reviewer for this suggestion and in alignment with a response to a similar point raised by reviewer 1 we added the following information regarding the inter-rater reliability; ‘In this study, individual scores only ranged between 2 (thin) and 3 (normal), with an overall inter-rater reliability of 77%.’ (line 146-147). As all individual scores given ranged only between a BCS of 2 and 3, we did not re-evaluated mismatching cases, but rather calculated an finally assigned the mean out of two individually given scores for an individual for a given month and hope the reviewer agrees on the suitability of this approach.
Line 126: Further clarification of how samples were collected is needed. Generally, sub samples of several bolus are taken to avoid hormone “hot spots”. Hormone metabolites are not evenly distributed across an entire faecal sample, especially animals that defecate in bolus form and where one bolus is a part of a whole. Sub-sampling several boluses to get a “homogenous” sub-sample is best practice. As I understand it, sub-samples were taken from only one bolus per elephant. If that is the case it should be clearly stated or otherwise clarified further.
We agree with the reviewer and apologise for this misunderstanding. For sample collection, faeces were indeed removed from the middle of some boli of a dropping to get a ‘homogenous’ sub-sample. We rephrased the respective statement for clarification (line 153 in the revised MS).
Was GPS location of the samples recorded? A map of the CFR area would be good to include with home range for each herd depicted (if known), with particular interest to their proximity to artificial water sources mentioned and human settlements.
We thank the reviewer for the question and suggestions. Unfortunately, it was not possible for us to record GPS location of the samples. But we hope that including additional information like ‘…The grazers tend to congregate on the short lawn grasslands, while the elephants move around the reserve in search of suitable forage patches” (line 127-129)’, helps to clarify to the readers that elephants did not access to human settlements and used the whole 11,000 ha of the reserve with no normal patterns of home range or riding behaviour displayed by, for instance elephant populations in unfenced reserves in the East Africa.
Table 1: Inclusion of number of faecal samples analyzed per individual is needed in the table for proper interpretation of the results given the low number of samples and the wide range of samples per individual (0-4). Indicating that some values reported are one-off samples while others are the median, but the title of the table says median for all which is not the case for the n=1 individuals.
We thank the reviewer for pointing that out. We adjusted the table heading accordingly and added missing detailed information (Table 1, line 191)
Additionally, adding an asterix or some other indicator of significance to table 1 between the April and June values would be helpful to the reader in interpreting the table. Not sure where I should add it.
Thanks for the suggestion, we now indicated statistical significant differences between groups in the revised table (Table 1, line 187).
Section 3.1 is repeated in the manuscript reviewed.
We apologize for this formatting mistake and deleted the repeated heading (line190-200 in the revised MS).
Line 188-189: It is known what these elephants eat? How many different vegetation types do they consume and do they consume agricultural crops? An additional clarifying statement regarding what is known about what they consume should be included, otherwise a statement that it is not known and they could be supplementing with agricultural crops should be mentioned.
Unfortunately, it is currently unknown what vegetation types these elephants actually consume. However, further studies on this particular aspect are currently in the planning stage by Wildlife and Ecological Investments (WEI), and thus future results will hopefully help us to answer this interesting question. To address the point of additional nutritional support, as pointed out above, we added respective information to clarify that the elephants could not access crops or were supplementary fed (line 254-255).
Lines 193-194: Need to temper this assertion as it could only be verified for six individuals based on very low number of samples. Something along the lines of: "for the individuals where matched values were available, median fGCM concentrations were 33% higher in June compared to April".
We adjusted the statement as suggested (line 209-210 in the revised MS).
Most the animals observed are females, and it is known that fGCM fluctuate with the ovarian cycle in elephants with higher concentrations occurring during the follicular phase (see Fanson et al. 2014; doi: 10.1530/EC-14-0025). April and June are far enough apart to potentially be in two different phases of the ovarian cycle and could explain the differences seen in the concentrations. Although knowledge of the ovarian cycle for each individual elephant is beyond the scope of this paper, it needs to be mentioned and discussed as possible explanation.
We thank the reviewer for pointing that possible explanation out and we adjusted the MS accordingly (lines 244247 in the revised MS)
Line198-199: the statement “our results showed no seasonal fluctuation in individual BCS…” is confusing because to my understanding samples and pictures were taken in the middle of the same rainy season (March – September 2018). Comparing to studies that looked at BCS across seasons is not appropriate for interpretation of the results.
We thank the reviewer for raising that point. We omitted the words “seasonal fluctuation” and in accordance to previous comments adjusted our emphasis in interpreting changes ‘across a season following severe drought’ instead of evaluating seasonal-related changes.
There’s are also no “seasonal related changes in BCS” (line 201-202) because there are no different seasons in this study design that I can find. Alternatively, explaining that within season there is no difference makes more sense, and that there is a need to investigate this relationship across seasons in future studies.
We agree with the reviewer and as pointed out above adjusted our emphasis in interpreting changes ‘across a season following severe drought’ instead of evaluating seasonal-related changes.
Cortisol also increases during musth in bulls, some incorporation of this in the discussion could be useful in explaining the results.
We deliberately have not referred to a potential explanation of elevated GC concentrations based on the dominant reproductive strategy of males (musth), as this relationship hasn’t been clarified completely, with a number of publications either didn’t demonstrate such a relationship or in fact show that respective fGCM concentrations are actually reduced during musth (e.g. Ghosal et al. 2013; Rasmussen et al. 2008, Ganswindt et al. 2005a, b, 2010).
Ghosal et al., 2013 PlosOne, 8(12): e84787
Rasmussen et al. 2008 Hormones and Behaviour, 54: 539-548
Ganswindt et al. 2010 Hormones and Behavior, 57: 506-510
Ganswindt et al. 2005 Hormones and Behavior, 47 (1): 83-91
Ganswindt et al. Physiological and Biochemical Zoology, 78 (4): 505-514
Lines 215-216: From the explanation of the lactating female with BCS 2.0 it does not seem to me that she is under nutritional stress. She is able to provide nutritional support for her calf and herself while increasing BCS in three months, she does not appear to be compromised in any way. Knowing which one she is in terms of fGCM would be useful in this discussion as cortisol is primarily a metabolic hormone. Discussion of Cortisol role in maintaining metabolic homeostasis during lactation is needed and would round out the discussion nicely. As much broad perspectives as needed should be taken given the limited data available.
We thank the reviewer for bringing this to our attention, and we added respective information to the manuscript. ‘… Unfortunately, we did not manage to get a fGCM value for Female β2 for June to be compared with the average fGCM concentration determined for this female for April. Therefore any discussion about a potential modulating effect of elevated glucocorticoid concentrations to support existing energy demands, as seen in other studies [48,49], would be speculative’ (line 236-240 in the revised MS)
Lines 222-223: How much difference was there in rain during the same season to account for this interpretation? Especially given that artificial water sources are also present in this area. This is over interpreting data in my opinion and should be re-examined.
We added information regarding rainfall for the season studied as well as for the period prior to the drought (line 117-119) and underlined the fact that the extensive drought registered in the region between 2015 and 2018 impacted on the vegetation. In the revised version of the manuscript we say ‘…While the artificial waterholes maintained water for drinking, the perennial lawn grasslands were severely affected by the drought and remained brown throughout the study period, and the elephants did not have access to supplementary forage.’ (line 250-254 in the revised MS). We hope this helps to alleviate the concern.
Lines 226: Discussion of an exceptionally dry conditions over the winter rainfall season needs to include discussion of the artificial water sources that are present in this area. How often are the elephants known to visit these sites? Wouldn’t the dry conditions be offset by the artificial water sources? How much impact do they have on the ecology of the area?
As just mentioned, we added some information regarding rainfall for the season studied and before the drought (line 117-119) as well as underlined the fact that the extensive drought registered in the region between 2015 and 2018 impacted the vegetation. Although artificial waterholes maintained water for drinking, they did not offset any stress on the impacted vegetation.
Lines 233-236: Discussion of the other potential sources for an increase in fGCM should include the possibility of increased activity of individual B1. With more activity cortisol will increase and would not necessarily be the result of “stress induced” increases in the measured fGCM. An animal that can modulate cortisol secretion to better cope in a changing environment is able to adapt and respond to those changes and would be maintaining homeostasis.
We acknowledge the reviewers view and agree that numerous intrinsic and extrinsic factors might affect the hypothalamic pituitary-adrenal axis, resulting in an increase in glucocorticoid production as a stress-defining countermeasure of restoring homeostasis. As pointed out earlier, we unfortunately don’t have any information regarding movement or activity pattern and thus can’t reliably link that aspect to the fGCM pattern seen e.g. in bull β1. However, to broaden the original ‘stress focused’ perspective we extended the final statement of the paragraph now ending ‘…, longitudinal physiological data in combination with BCS together with additional comprehensive measurements of ecological conditions, could also allow a better understanding of the overall coping mechanisms these elephants use to thrive in such environments.’ (line 271 -273)
A broader view is needed in this discussion overall. The animals appear to be coping well, which is great, more exploration of why that would be and what facilities their persistence in this area would be good alternatives to explore rather then a singular and very simplistic and outdated “stress” perspective.
In line with the comment above, we broaden the original ‘stress focused’ perspective by extending the final statement of the paragraph (line 266 -273).
Lines 243-244: Hard to compare free roaming animals to animals held in zoos that are generally overfed and have reduced activity in comparison, with very different environmental and psychological stressors to contend with. If you need examples of other studies that looked at BCS and fGCM there is a wealth of information available in other species such as bears (omnivores and I think still appropriate to use) and ungulates. Those would be better comparisons for a general discussion of how associations between BCS and fGCM can be used to optimize conservation strategies and the challenges with making these associations in the face of complex dietary patterns and seasonal variations.
We principally agree with the reviewer, and therefore referring now to studies wild rabbit and stellar sea lions as well as African and Asian elephants [24,25,], actually demonstrating a negative relationship between fGCM concentrations and BCS (line 274-278 in the revised Ms). The reason for also presenting the study by Kumar and colleagues was, that despite the fact that those elephant are held in zoos with a presumably suboptimal diet, reduced activity in comparison, and very different environmental and psychological stressors to contend with, they also didn’t found a respective correlation, which we think is worthwhile noting.
Lines 248: typo – correct to moths to months. And wasn’t it three months of study?
Corrected - Thanks for pointing that out (line 281 in the revised MS). The months of study were two, April and June 2018, but total duration obviously stretched over three months.